# First Report of Genetic Variability of *Erysipelothrix* sp. Strain 2 in Turkeys Associated to Vero Cells Morphometric Alteration

**DOI:** 10.3390/pathogens10020141

**Published:** 2021-02-01

**Authors:** Thais Fernanda Martins dos Reis, Patrícia Giovana Hoepers, Phelipe Augusto Borba Martins Peres, Eliane Pereira Mendonça, Paula Fernanda de Sousa Braga, Marcelo Emilio Beletti, Daise Aparecida Rossi, Ana Laura Grazziotin, Luiz Ricardo Goulart, Belchiolina Beatriz Fonseca

**Affiliations:** 1School of Veterinary Medicine, Federal University of Uberlandia, Uberlandia-MG CEP 38400-902, Brazil; thais_koro@hotmail.com (T.F.M.d.R.); patriciag.hoepers@gmail.com (P.G.H.); lipe-peres1@hotmail.com (P.A.B.M.P.); eliane_vet@yahoo.com.br (E.P.M.); paulinhafsb@gmail.com (P.F.d.S.B.); daise.rossi@ufu.br (D.A.R.); analauragrazziotin@gmail.com (A.L.G.); 2Institute of Biomedical Sciences, Federal University of Uberlandia, Uberlandia-MG CEP 38400-902, Brazil; mebeletti@ufu.br; 3Institute of Biotechnology, Federal University of Uberlândia, Uberlandia-MG CEP 38400-902, Brazil; lrgoulart@ufu.br

**Keywords:** *Erysipelas*, injuries, transmission electron microscopy, PFGE, apoptosis

## Abstract

Erysipelas is a disease caused by the *Erysipelothrix* genus, whose main species is the *E. rhusiopathiae*, the causative agent of animal erysipelas and human erysipeloid. We isolated *Erysipelothrix* sp. strain 2 (ES2) from turkey’s organs during an outbreak in Brazilian commercial and breeder flocks with sepsis and high mortality levels. We studied 18 flocks, accounting for 182 samples, being eight flocks (84 samples) as ES2 positive with individuals demonstrating clinical symptoms and high mortality. We obtained the genetic variability of 19 samples with PFGE and found two clones, both from the same flock but different samples, and two clusters. Interestingly, we found 15 strains with high genetic variability among and within flocks. We have found a positive association between the proximity of ES2 positive turkey flocks and commercial swine sites through epidemiological analysis. We infected Vero cells with two different isolates and three distinct concentrations of ES2. After performing the morphometry, we recorded enlargement of the nucleus and nucleolus. Moreover, we performed fluorescence assays that resulted in apoptotic and necrotic cells. We demonstrated that ES2 could multiply in the extracellular medium and invade and survive inside Vero cells. For the first time, our finds show that ES2 may have similar behavior as *E. rhusiopathiae* as a facultative intracellular microorganism, which may represent a hazard for humans.

## 1. Introduction

Erysipelas is a worldwide disease that affects several vertebrate species such as humans, domestic and wild animals [1]. Although other species have been reported, *Erysipelothrix rhusiopathiae* is the leading cause of illness with an increasing number of erysipelas cases in swine, chickens, and humans [2,3,4,5]. The cutaneous form (erysipeloid) of *E. rhusiopathiae* is the most common infection in humans [6]; it is often occupationally related, occurring mostly in people who work closely with contaminated animals, their products, wastes, and soil, such as butchers, abattoir workers, veterinarians, farmers, and fishers [7]. The invasive disease is rare and associated with infective endocarditis, with high mortality demanding prompt diagnosis and treatment [8]. In turkeys, *Erysipelothrix* spp. causes an acute infection, septicemia, reduced feed intake, carcass condemnation, mortality, and male infertility [1].

*Erysipelothrix* spp. are Gram-positive and rod-shaped bacteria [9], belonging to the Erysipelotrichaceae family (Phylum Firmicutes; class Erysipelotrichia; order Reysipelotrichales) [10]. There are eight known species, namely: *E. rhusiopathiae*, *E. tonsillarum*, *E. inopinata, Erysipelothrix* sp. strain 1, *Erysipelothrix* sp. strain 2 (ES2), *Erysipelothrix* sp. strain 3 [10,11,12], as well as the last identified *Erysipelothrix larvae* sp. nov. [13] and *Erysipelothrix piscisicarius* sp. nov [14] 

Before ES2 was identified as *E. rhusiopathiae* serotype 18, this serotype and other *E. rhusiopatiae* serotypes were highly virulent for mice and pathogenic for swine, causing localized urticaria [11]. Erysipelas is an emerging disease in poultry-producing farms [15], having outbreaks reported in turkeys being caused by *E. rhusiopathiae* in previous studies. There is still a limited understanding of the disease caused by ES2 on a global scale for both animals and humans. Additionally, *Erysipelothrix* spp. could be misidentified during outbreaks due to significant biochemical and phenotypic similarities between *E. rhusiopathiae* and ES2.

During disease outbreaks, correctly identifying genetic differences among isolates is crucial to better understanding the disease epidemiology [16]. Pulsed-field gel electrophoresis (PFGE) plays a key role in outbreak-associated isolates and their genomic variability identification, and thus, PFGE provides an excellent epidemiological tool. Moreover, the geographic recognition of pathogen’s hosts is useful in attributing the epidemiological identification of the species [17]. Therefore, the combination of PFGE with georeferenced occurrences certainly enrich epidemiological studies. Knowledge regarding pathogens infections outcomes in cellular phenotypic is imperative for pathogenicity and immunity studies and enhance prevention and control actions. Cellular cytotoxicity and life-history traits evaluation of newly discovered species or understudied bacteria is essential information before inoculation studies in animals. As an example, one may use morphometry and apoptosis analyses to assess cell alteration induced by pathogens. 

Previously we reported an outbreak caused by ES2 [18], which had not been identified in turkeys so far. In the present study, we aim to access the genetic variability of ES2 isolates, the epidemiological relationship with swine properties, the pathogenic outcomes of infection with the bacteria in Vero cells, and phenotypic traits of ES2 through transmission electronic microscopy (TEM). 

## 2. Materials and Methods

### 2.1. Samples

From January 2014–2015, an outbreak causing sepsis and mortality in commercial turkey flocks affected a large Brazilian poultry-producing company. We identified the cause being *Erysieplothrix* sp. strain 2 (ES2) infection [18] in organ samples collected from sick birds. Therefore, for the current study, we selected a few flocks for analyses during the outbreak (from December 2014 and March 2015).

The study design consisted of 35 commercial turkey flocks, being 34 male Nicholas lineages, ranging from 36-day-old individuals until the slaughtering day. Additionally, we used one Nicholas turkey lineage belonging to breeder flocks with and without clinical disease. We necropsied turkeys up to one hour after their natural death and collected the following organs: liver, lung, spleen, kidneys, trachea, intestine, and heart. We put the same organs from two to three different birds in a sterile pack and sent the samples to the laboratory for isolation and identification following the protocol described by Hoepers et al. (2019). Although we analyzed newly dead birds, we informed the Committee on Ethics in the Use of Animals (CEUA) of the Federal University of Uberlandia upon the advice number A004/19.

In order to avoid selection bias, we selected flocks with similar rearing conditions (house structure, food, and water management). Flocks were also negative to veterinary interest microorganisms, which may cause high mortality and septicemia symptoms when isolated or in co-infections (*Pasteurella multocida*, *Escherichia coli*, *Erysipelothrix rhusiopathiae*, *Salmonella* spp., *Streptococcus* spp., *Staphylococcus* spp., *Aspergillus* spp., and New Castle disease) [18]. 

After laboratory analysis, we selected 18 flocks, distinct from the ones described by Hoepers et al. (2019), being eight with and ten without clinical signs. In total, 182 samples were studied, being 84 from positive flocks and 98 from negative flocks (Appendix A). We isolated ES2 from 33 samples from the 84 collected from the positive flocks (Appendix A). In the positive flocks, we isolated just ES2 but not other species of *Erysieplothrix* spp. In the ES2 positive group, seven were commercial turkey flocks and one from a breeder’s turkey flock. The final average mortality was 13.11% for ES2-positive flocks and 6.48% for ES2-negative flocks (Appendix A). The average age of positive birds for ES2 was 120.6 days (Appendix A).

### 2.2. Identification of Erysipelothrix spp.

We performed the identification of *Erisypelothrix* spp following the protocol described by Hoepes et al. (2019) [18] at the Animal Health Laboratory. We inoculated the samples on sheep blood agar (OXOID) and MacConkey agar (OXOID) and incubated them aerobically at 37 °C from 18 to 24 h. We picked typical *Erysipelothrix* spp colonies and submitted them to Taqman real-time PCR for differentiation among *E. rhusiopathiae, E. Tonsillarum*, and ES2. The DNA primer set probes are highly specific for detecting *E. rhusiopathiae, E. tonsillarum* and *Erysipelothrix* sp. strain 2, and we set up the reaction conditions according to Hoepers et al. (2019).

### 2.3. Transmission Electronic Microscopy (TEM)

We performed TEM at the Electron Microscopy Center from the Federal University of Uberlandia, Uberlandia, Minas Gerais, Brazil, to access bacterial ultrastructure of two isolates of ES2 (Est06 and Est07). We fixed the isolates and washed them using 0.1 M phosphate with buffer (pH 7.2), centrifuged at 2900 rpm for 20 min, and placed them on agar 1%. We counted the resultant material in cubes of approximately 3 mm^3^ and placed them in osmium tetroxide for one hour consisting of 1% 0.1 M phosphate buffer (pH 7.2), being the treatment of 1% osmium tetroxide plus and 1.25% potassium ferrocyanide for 30 min. We dehydrated the fragments by increasing the acetone concentrations, placed them in epon resin, and cut them with an ultramicrotome to obtain ultrathin cuts. We obtained ultrathin sections and contrasted them with uranyl acetate and lead nitrate on small nickel screens [19].

We analyzed the small screens in a transmission electron microscope (Hitachi HT—7700, Hitachi Ltd., Tokyo, Japan) with ESPRIT data acquisition and analysis software.

### 2.4. Pulsed Field Gel Electrophoresis (PFGE)

We evaluated the genetic similarity of 19 isolates of ES2 using pulsed-field gel electrophoresis following the protocol recommended by CDC (2013) [20] with modifications. We tested 19 isolates because others did not survive, as shown by previous conservation methodologies. We used the following isolates: Est02, Est03, Est04, Est06, Est07, Est08, Est09, Est11, Est12, Est14, Est16, Est17, Est18, Est23, Est24, Est26, Est28, Est29, and Est30. We cultivated the bacteria isolates in BHI (brain heart infusion) agar supplemented with 5% defibrinated lamb blood and incubated at 37 °C, and treated them with *SmaI* restriction enzyme (40 U/uL) to continue with the plug modeling. The isolated fragments resulted from pulsed-field electrophoresis, with the subsequent set of conditions: an initial switch time of 2.2 s and a final switch time of 64 s, with a gradient of 6 V/cm for 21 h. We separated DNA fragments with 1% agarose gel (SeaKem Gold) in 0.5× TBE buffer belonging to CHEF-DR^®^ III Pulsed Field Electrophoresis Systems (Bio-Rad) and buffered temperature of 14 °C. We utilized *Salmonella* Braenderup as the internal reaction control.

We stained gels with ethidium bromide, photographed under UV light, and built a dendrogram using Bionumerics 7.6. We obtained the final pattern with the Dice similarity coefficient with a 0.5% tolerance and performed the dendrogram model using the UPGMA (unweighted pair group method with arithmetic mean) analysis method.

### 2.5. Epidemiological and Statistical Analysis

We compared data from ES2 positives and negatives flocks. Despite all properties having acceptable biosafety standards, only the breeding farm had daily disinfection tools and mandatory showers for staff and visitors. The longer distance among turkey farms was up to 110 km. We considered 7.0 km as a close distance between turkey and swine farms and used only commercial swine farms from the same company for the proximity analysis due to a lack of information about other properties. We made a map using Google Maps and QGIS to display better the distribution of the 19 studied isolates (Appendix A) in the properties and the proximity with swine herds. We analyzed data of proximity between swine farms with positive samples of ES2 by association analyses.

### 2.6. Cellular Morphometry

We used diverse concentrations of the bacterial inoculum to test the effects of ES2 in Vero cell morphology. We seeded Vero cells in DMEM (Dulbecco’s Modified Eagle Medium) enriched with 5% bovine fetal serum in circular glass slides with coverslips and placed them in a six-well plate at a density of 1.68 × 10^4^ cells/well and incubated for 24 h. We measured cell growth within each sample and inoculated in triplicates of two ES2 isolates (Est06 and Est07), using the following bacteria inoculum: D0: 1 Log CFU/well; D2: 2 Log CFU/well; D6: 6 Log CFU/well and NC: Negative control treated with sterile PBS. After a period of four and 24 h post-inoculation (p.i.), we removed the coverslips and fixed them in formalin 10%, washed them in PBS, stained them with panoptic dye, and, lastly, placed them on slides with Entellan (Merk).

We carried the visualization and photographic documentation under a type Olympus BX 40 camera-coupled microscope (Olympus camera, 200, Olympus Optical Co., Ltd., Tokyo, Japan) under 100× magnification, using the Data Translation 3153 software. We performed the image editing, processing, and analyses with the Image J program, version 1.51, 2017, evaluating eight different fields per slide, totaling 64 cells per slide. We measured the following parameters: nucleus area, nucleus perimeter, nucleolus area, and nucleolus perimeter.

### 2.7. Apoptosis

We seeded Vero cells in DMEM enriched with 5% bovine fetal serum in circular glass slides and placed them in a 6-well plate at a density of 2 × 10^4^ cells/well and incubated for 48 h at 37 °C and 5% CO_2_ atmosphere when cells were confluent. We then performed an inoculation in triplicate: (i) ES2 Est06 using 3 log CFU/well, (ii) ES2 Est07 using 3 log CFU/well, (iii) 3 log CFU/well *Lactobacillus* spp (LB) isolated from Probiotic milk beverage fermented (CALU) in MRS agar (deMan, Rogosa, and Sharpe) (OXOID), and (iv) NC: Negative control treated with sterile PBS. After a period of four and 24 h, we washed the cells three times with PBS, treated them with Yo Pro 01(YP) and propidium iodate (PI) in the concentration of 1:1000 each (Invitrogen) for 30 min at room temperature. We then washed them three times and fixed them with 4% formalin for 10 min. After that, we treated cells with Hoechst (Sigma) to mark the cell DNA. We washed slides and arranged them in an anti-fading ProLong (Invitrogen). We analyzed counts under a fluorescence microscope (EVOS FL Cell Imaging System, Life Technologies Corporation, Carlsbad, California, USA), evaluating five different fields per slide.

### 2.8. Invasion Test

We performed a bacterial invasion assay after 24 h and quantified the extracellular bacteria. We then extracted 100 uL of the supernatant, and after serial dilutions, we accounted for blood agar or MRS agar for the presence of ES2 and *Lactobacillus* spp., respectively. We washed the cells four times with PBS and performed another bacterial count in the fourth wash. We treated cells with trypsin to remove the plate’s cells, then centrifuged them (1500 rpm/10 min), treated them with triton 1% for cell membrane disruption and release of intracellular bacteria. We finally washed four times (1500 rpm/10 min), counting the bacterial for the last time.

### 2.9. Statistical Analysis

We analyzed data of proximity between swine farms with positive samples of ES2 by association analyses (chi-square and Fisher’s test). Additionally, we used chi-square and Fisher’s test, followed by binomial between two proportions for apoptosis and necrosis analyses. We used ANOVA to test for differences in cells’ morphology, with a 95% of confidence level, and performed all analyses in GraphPad Prism 7.04.

## 3. Results

### 3.1. Electron Microscopy of ES2

The photomicrographs identified two different cutting positions for ES2: longitudinal section in rod format (Figure 1A,B) and cross-section (Figure 1A–D). The bacterial wall can be visualized (Figure 1A,C,D) as well as the cytoplasmic membrane (Figure 1B–D). We measured the bacillary form mean and obtained 2.16 µm × 0.55 µm.

### 3.2. Epidemiologic Analysis Relative to PFGE

We found a positive association between positive-ES2 flocks and proximity to swine farms with an odds ratio (OR) of 21.55 (Appendix A). Turkey farms identified as D, E, G, and H were close to swine farms; some were as close as 1.7 km (Figure 2 and Appendix A). We found high genetic variability among the 19 ES2 isolates tested on PFGE (Figure 2) with different banding patterns observed between isolates; we show these results in the Appendix A. There were only two clones (100% similarity) and two clusters (>90% similarity). We isolated clones Est02 and Est03 from birds in farm A, respectively from the pooled samples of heart and spleen, clones Est14 and Est16 from birds in farm E, respectively isolated from the pooled samples of lungs and heart (Figure 2). We observed that isolates Est04 and Est07 were located in the same cluster; interestingly, we isolated Est04 from turkeys’ lungs in farm A, whereas Est07 was from turkeys’ kidneys in farm B. We isolated Est17 and Est24, grouped in another cluster from livers of turkeys’ in farm C, whereas Est24 from turkeys’ spleens in farm H.

Although we identified clones and clusters, we observed high variability among the isolates studied even in samples from the same flocks. We obtained the isolates Est08, Est09, and Est11 from the same flock in farm D and found only 44.8% similarity. Moreover, we found that Est012 is different from the clones Est014 and Est016 (56.3% similarity), even that we isolated them from farm E. We obtained similar high genetic variability results with isolates Est24, Est26, Est28, Est29, and Est3 obtained from the same flock in farm H and isolates Est02 and Est04, and Est06 and Est17 isolated from the same turkeys flock but from different organs (Figure 2).

### 3.3. Morphometry in Vero Cells

To verify cellular morphometry changes, we infected Vero cells with Est06 and Est07. After four and 24 h, for all samples of the evaluated ES2, we observed a significant increase of all measured parameters (nucleus area, nucleus perimeter, nucleolus area, and nucleolus perimeter) in the Vero cells treated with Est06 and Est07 isolates (Figure 3, Figure 4 and Figure 5) (Appendix A).

For the treatment with Est06, the nucleus area was larger at four hours p.i for D2, the intermediary quantity of bacteria (2log CFU/well), than for D0 (1log cfu/well) and D6 (6log CFU/well). At 24 h p.i. D0 and D2 equally led to bigger nucleus than D6. For the nucleus perimeter, as expected, we observed the same pattern (Figure 3A,B). For the nucleolus area, the treatment D0 led to larger areas at both four and 24 h p.i., for the nucleolus perimeter, we observed similar results (Figure 3C,D). For the treatment with Est07, D0 caused the most relevant impact on nucleus size and perimeter at four- and 24 h p.i. (Figure 4A,B). For the nucleolus area, at four hours p.i., we found the largest area associated with D0. At 24 h p.i., we observed the same average size for both D0 and D2. The results of the nucleolus perimeter were similar (Figure 4C,D).

### 3.4. Apoptosis

We found a higher percentage of cells labelled with YP in the Est06 and Est07 treated groups than those treated with LB or without any treatment. Besides, we verified that at four hours p.i., the Est07 group showed a higher percentage of cells labelled with YP than other groups and that cells treated with Est06 and Est07 had a higher number of PI-labelled cells than controls, except for Est07 at 24-h p.i. (Figure 5 and Figure 6).

### 3.5. Bacterial Multiplication

After 24 h, ES2 multiplied in the extracellular medium since we infected 3 logs CFU/well, and after 24 h, we obtained 7.72 log CFU/well of Est06 and 7.11 log CFU/well of Est07 in the extracellular medium of the cells infected by these isolates (Table 1). We did not find *Lactobacillus* spp. in the Vero cell culture medium. We washed the cells four times with PBS and performed another bacterial counting in the fourth wash (with the intact cell) (Table 1). We found no difference between the counts of Est06 and Est07 isolates (Table 1). We treated cells with trypsin for the removal of the plate’s cells, then centrifuged them (1500 rpm/10 min), treated them with triazole for cell membrane disruption and releasing of intracellular bacteria and then, washed them four times (1500 rpm/10 min), counting the bacteria for the last time. We found a statistically significant difference between the number of bacteria after the integrated cells wash and the number of bacteria after the triton membrane dissolution (Table 1). This indicates that the bacteria remained in the intracellular environment after 24 h post-inoculation.

## 4. Discussion

We measured and revealed the ultrastructural characteristics of ES2 in TEM images for the first time. We found the expected typical rod-shaped (longitudinal section) and circular-shaped bacteria (cross-section) and 2.16 × 0.55 µm. In photomicrograph, we visualized the bacteria’s wall and cellular membrane. Boerner et al. (2004) [21], used TEM to evaluate *E. rhusiopathiae* isolated from a blue penguin. They found slender bacilli, with osmiophilic cell walls, without other wrinkled membranes. We obtained the first ES2 electron micrograph, and such images are imperative for future studies to assess the association between pathogen and host cells. However, it is necessary to apply other different sample processing TEM methodologies to better characterize the bacterial ultrastructure and the pathogen-host cell interaction in future studies.

We found high genetic variability in the ES2 isolates in the PFGE analysis (Figure 2 and Appendix A). In another study, we sequenced the strains Est06 and Est07 and these strains were different as well, showing that the outbreak was not caused by one only clone [22]. Even though we expected to find isolates belonging to the same cluster infecting the same flock, we hypothesize that the variability found could be explained by the diverse possibilities of contamination and re-contamination of the properties’ environment and further on the birds. *Erysipelothrix* spp. have been isolated from environmental sources, such as dust, water nipples, and manure [23,24], suggesting that the bacteria may be widely spread within the environment. In a study performed by Eriksson et al. (2014) [24], *E. rhusiopathiae* was isolated in contaminated dust samples from exhaust fans, which led the authors to speculate about the possibility of the microorganism leaving commercial turkeys environment and disseminate to wild animals.

Moreover, the bacteria dissemination may occur in turkey flocks by human contact, contaminated manure, boots, equipment and feed, rodents, and wild animals. Although other animals such as chickens and ducks are also susceptible to *Erysipelothrix* spp, swine are the domestic animals most affected by *Erysipelothrix* spp., being its most important reservoir [25]. Swine can be *E. rhusiopathiae* carriers (which means that this agent may be isolated from different places within the swine farms, such as drinking fountains, walls, feed, and water [23]. Similar serotypes of *E. rhusiopathiae* can be found in swine and turkey flocks as well [26].

Since we found an association of ES2 positive turkey flocks and proximity to swine farms (1.7 km–7 km), we speculate if the swine could be a constant source of contamination for turkey’s flocks nearby, contributing to the isolates’ genetic variability. As we showed in Figure 2, one facility for swine semen collection was quite near property D, and in property H, the same person also owned a swine property. The same feed mill produced feed for the turkeys and swine; besides, the trucks that carried the feed could also play a role in the contamination between swine and turkey properties. We also highlight an intense concentration of swine, chicken, and turkeys’ properties in the region where the outbreak occurred, which may corroborate the spread of distinct ES2 to the same turkey’s flock. In this study, we did not track the sources of contamination. Thus, further investigation is necessary to verify the hypothesis presented above. Our research’s critical bias is the lack of information about the backyard pigs and chickens, which could certainly play a role in the source of contamination and, thus, genetic diversity of isolates.

The turkeys’ infection with ES2 led to clinical signs and macroscopic lesions similar to the ones caused by *E. rhusiopathiae* [18], which is the principal agent of erysipelas in turkeys [1]. We observed an increase in the nucleus and the nucleolus area and perimeter at four and 24 h after inoculation in Vero cells with ES2 (Figure 3, Figure 4 and Figure 5). However, we could not study which virulence proteins could generate this cellular alteration, such as alpha-hemolysin [27] and pyrogenic exotoxin B [28] that can induce apoptosis in Gram-positive bacteria. Ogawa et al. (2011) [29] performed the first complete genome analysis of *E. rhusiopathiae* and found that the bacteria display several wall-associated virulence factors, such as capsule and adhesins. Additionally, its genome encodes nine antioxidant factors and nine phospholipases, which allow the survival in phagocytes. 

Wang et al. (2017) [30] studied several cell wall-associated proteins from high and low virulence *E. rhusiopathiae* isolates. They found that the proteins with higher abundance in high-virulent strains were mainly ABC transporter proteins and adhesion proteins. These proteins may be related to bacterial virulence. More recently, Zhu et al. (2018) [31] demonstrate the pathogenic role of HP0728 (ERH_0728) and HP1472 (ERH_1472), both internalin-like proteins, as *E. rhusiopathiae* virulence factors. Further studies are necessary to verify if ES2 shares the same pathogenic factors with *E. rhusiopathiae*.

In the current study, fewer inoculated bacteria increased cellular parameters, on average, compared to the highest number of ES2. We hypothesize that this may result from fewer bacteria reaching more culture medium and infecting more free cells. 

An increase in nucleus and nucleolus dimensions led us to evaluate cell apoptotic index (Figure 5 and Figure 6) using YP. Therefore, cells marked only with YP are cells in earlier stages of apoptosis still metabolically active cells, the DNA fragmentation has not yet occurred but with compromised plasma membranes [32]. In contrast, PI detects cells that are already dead due to the delay in PI penetration compared to YP. If we consider the clinical form of erysipelas in turkeys, the possible apoptosis caused by such bacteria may be vital to understanding ES2 pathogenesis. A previous study has shown that *E. rhusiopathiae* can survive and replicate within macrophages and that reduced production of oxidative metabolites may play a role in this occurrence [33]. Nevertheless, the precise mechanisms that allow the bacteria to survive within the cell are still unknown [33].

During the outbreak, turkeys died from septicemia (with organ damage) or even due to acute mortality. Thus, the characteristic of acute disease infers that bacteria may somehow surpass the host’s immune system. On the other hand, tissue injuries caused by bacterial infection may be significant to inflammation [34]. Necrosis activates immune system cells and inflammation response [34,35], and in some situations, apoptotic cells can be pro-inflammatory [36]. Such events may explain the inflammatory lesions found in some animals. We suggest that cellular survival and apoptosis demonstrated in the present research may be an essential mechanism for ES2 virulence. Pathogens have developed intracellular survival mechanisms that may include or be only apoptosis to persist in the host’s cells [37,38,39]. Erysipelas led to sepsis in turkeys, with few individuals dying from acute mortality. Therefore, apoptosis may be harmful to the host and can be associated with enhanced bacteria virulence potential. Such a mechanism is considered a non-inducing inflammatory response [34]. 

We found a different percentage of apoptotic and necrotic cells infected with isolates 6 and 7. The number of necrotic Vero cells infected by Est06 was higher and occurred earlier than strain Est07 (Figure 5). Also, we observed a higher percentage of apoptotic cells four hours post-infection when comparing Est07 to Est06 (Figure 5). We attribute such differences to the genetic variability between the two isolates, shown in the PFGE, which led to pathogenicity differences. Interestingly, the Est 06 and Est07 were whole sequenced and they were different strains [22].

There is no information regarding ES2 life-history traits in cells. However, Días-Delgado et al., (2011) [40] found from 30 to 75% of *E. rhusiopathiae* intracellularly and extracellularly in the liver and kidney of a free-ranging Atlantic spotted dolphin with acute septicemia. *E. rhusiopathiae* is a facultative intracellular pathogen and can survive inside polymorphonuclear leukocytes and macrophages [33]. Our findings show that ES2 also survives and multiplicate in the extracellular medium and colonizes the Vero cells. The ability of ES2 to live in the intracellular environment is an excellent strategy for surpassing the host’s immune system [37].

In the present study, we showed that ES2 has a similar behavior as *E. rhusiopathiae* to thrive in the intracellular and extracellular environment. We highlighted the importance of the pathogen for poultry by studying its genetic diversity. Using Vero cells, we showed that apoptosis could play a significant role in erysipelas’ pathogenesis. We suggest that further studies to be carried to explain how the pathogen acts in the host organism to cause disease, the genes associated with high pathogenicity, and the epidemiological factors in turkey flocks that could have induced the genetic diversity observed.

Although there is treatment for ES2 with antibiotics, being the penicillin, the treatment of choice [1], the disease has a fast course, and the knowledge of the cellular mechanism and epidemiology of the disease is imperative for studies of prevention methods and development of other treatment. In particular when we consider that ES2 is a little-studied species.

Moreover, laboratories that perform human and animal diagnosis should implement the molecular diagnostic tools that differentiate ES2 and *E. rhusiopathiae.* This is paramount because we still lack information about this pathogen for other species rather than turkeys. Vero cells’ infection showed by us in this study indicates that the bacteria may, like *E. rhusiopathiae*, be a hazard for mammals, including humans, mainly workers in close contact with animals and its products. Even though the cutaneous form represents most human cases, the high mortality associated with the infective disease demands accurate diagnosis and treatment [8]. Thus, this study arouses the need for further studies on the zoonotic potential of ES2.

## 5. Conclusions

Isolates of ES2 isolated from ill turkeys with high variability can have high genetic variability. Additionally, this bacterium leads to cellular apoptosis with both cellular end extra cellular survive in mammals cells. 

## Figures and Tables

**Figure 1 pathogens-10-00141-f001:**
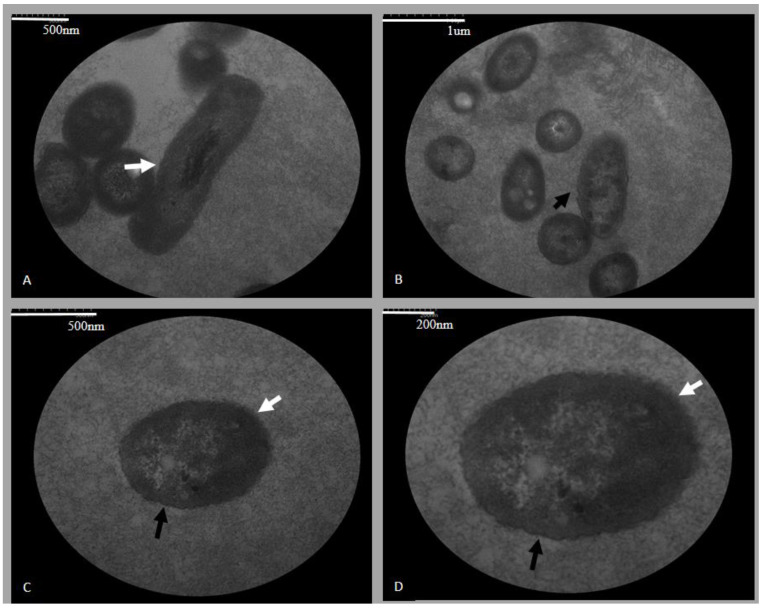
ES2 photomicrographs in different cutting positions. In (**A**,**B**), the bacterium format in longitudinal and cross section. In (**A**,**C**,**D**), part of the bacterial wall can be visualized (white arrow). In (**B**–**D**) cytoplasmic membrane can be visualized (black arrow).

**Figure 2 pathogens-10-00141-f002:**
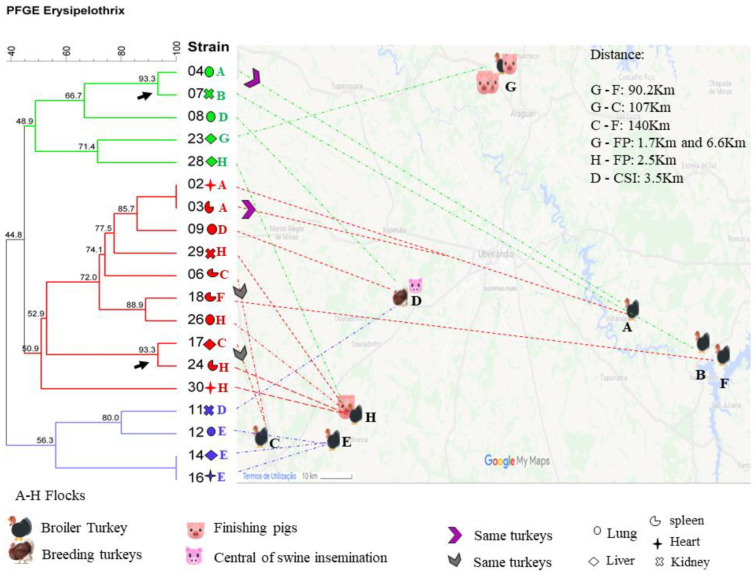
Georeferenced location of positive-ES2 turkeys’ flocks and swine farms, isolated organ, and the association with the PFGE. The black arrows show the clusters.

**Figure 3 pathogens-10-00141-f003:**
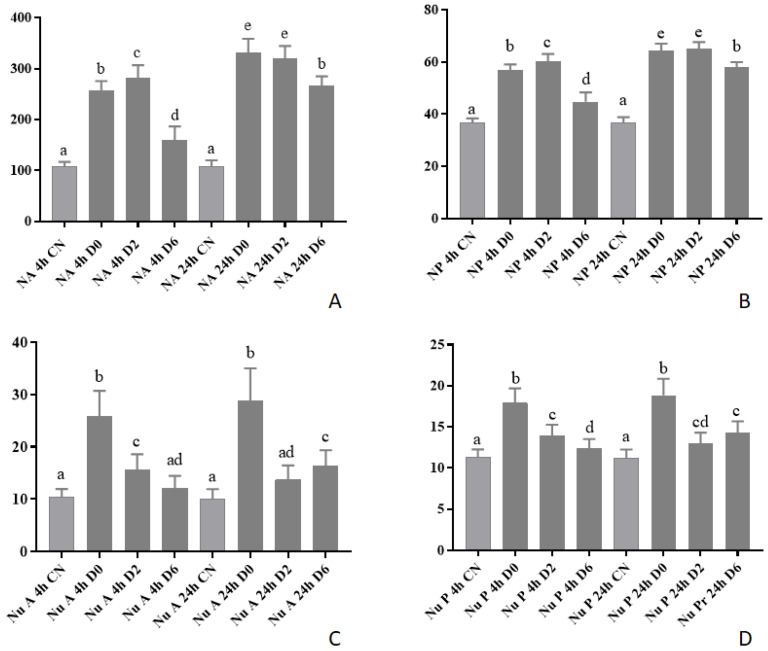
Vero cells morphometry at four and 24 h p.i. with Est06 within different concentrations. (**A**) NA: Nucleus area; (**B**) NP: Nucleus perimeter; (**C**) Nu A: Nucleolus area; (**D**) Nu P: Nucleolus perimeter (uM). Different letters in the same column represent the statistical difference (*p* < 0.05).

**Figure 4 pathogens-10-00141-f004:**
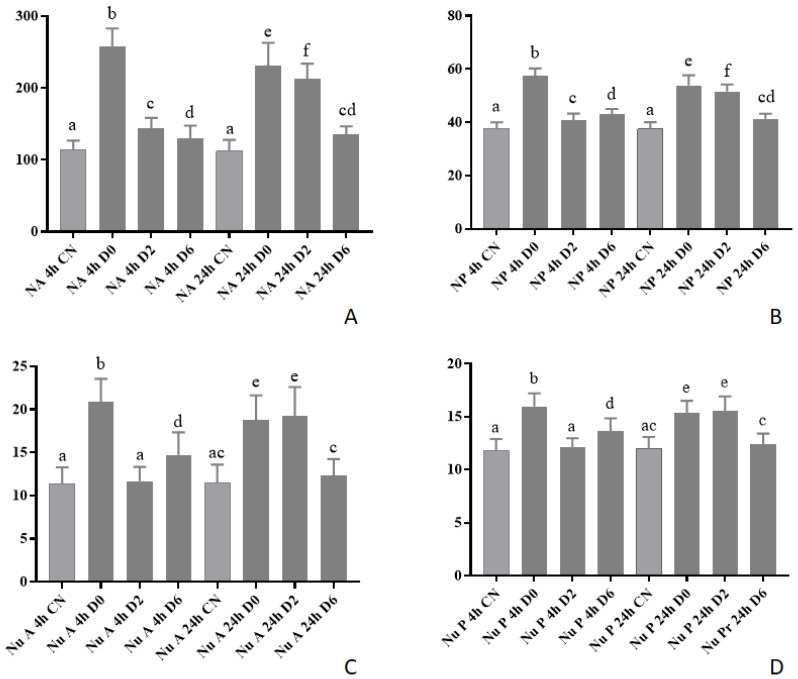
Vero cells morphometry at four and 24 h p.i. with Est07 within different concentrations. (**A**) NA: Nucleus area; (**B**) NP: Nucleus perimeter; (**C**) Nu A: Nucleolus Area; (**D**) Nu P: Nucleolus perimeter (uM). Different letters in the same column represent the statistical difference (*p* < 0.05).

**Figure 5 pathogens-10-00141-f005:**
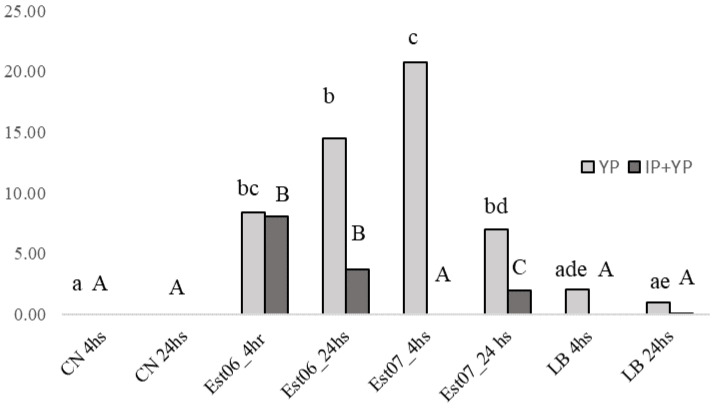
Percentage of cells labelled with YP and IP at four and 24 h p.i. Different lowercase letters indicate statistical differences for YP markings. Diverse capital letters indicate statistical differences for markings with YP and IP.

**Figure 6 pathogens-10-00141-f006:**
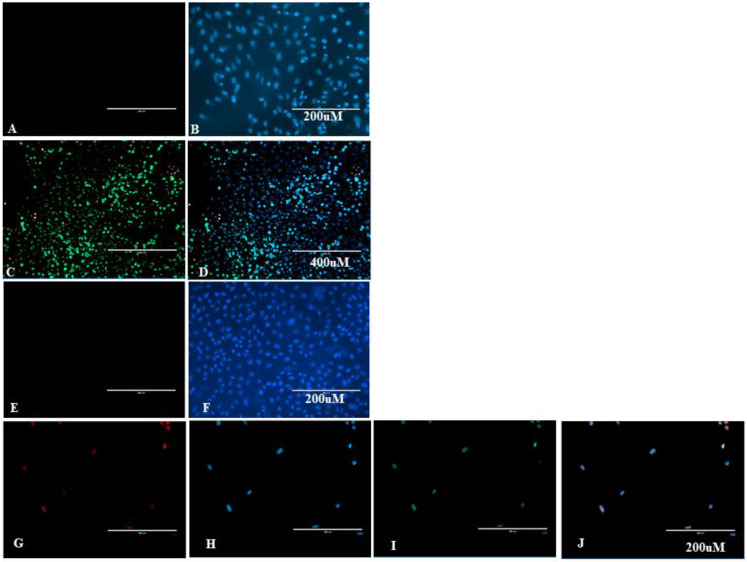
Fluorescence image of YP, PI, and Hoechst-labelled Vero cells. Scale bar is 200 µm or 400 µm. (**A**,**B**) Cells from the *Lactobacillus* spp. control group at four hours period p.i. showing only the Hoechst-labelled nuclei. (**B**) Cells were not marked with PI and YP (**A**); (**C**,**D**) Cells from the Est06 group at four hours period p.i. showing IP and YP-labelled (**C**) and YP, PI Hoechst-labelled nuclei (**D**); (**E**,**F**) Cells from the Negative control at 24 h period p.i. showing only the Hoechst-labelled nuclei (**F**). Cells were not marked with PI and YP (**E**). (**G**–**J**) Cells from the Est07 24 at hours period p.i. showing PI-labelled nuclei (**G**), Hoechst-labelled (**H**), YP-labelled (**I**), and all markers (**J**).

**Table 1 pathogens-10-00141-t001:** The average number of ES2 and LB (log CFU/well) in culture medium after 24 h and after several washes and triton treatment.

	Est06	Est07	LB
Initial inoculum	3.00 a	3.00 a	3.00
Inside the cell culture medium	7.22 b A	7.11 b A	0
After four washes	2.89 a B	2.81 a B	0
After treatment with triton	3.13 a C	3.23 a C	0

Lowercase letters should be interpreted concerning the initial inoculum. Uppercase letters represent the evaluation against bacterial count in the cell culture medium after 24 h of inoculation (extracellular bacteria). Different letters (lower- or uppercase) indicate the statistical differences.

## Data Availability

Data is contained within the article or Appendix A.

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
