# Peer review of "First Report of Genetic Variability of Erysipelothrix sp. Strain 2 in Turkeys Associated to Vero Cells Morphometric Alteration"

_pathogens, 2021, doi:10.3390/pathogens10020141_

Round 1
Reviewer 1 Report
Overall: This manuscript is generally well-written and provides new information on ES2. The information presented in the manuscript provides a basis for further studies on this pathogen, which may emerge as an important one affecting turkeys and other poultry.
Introduction:
Line 52: First use of abbreviation ES2 but no previous explanation in parentheses.
Line 72: Delete extra space in front of ES2.
Line 73-74: ES2 was suddenly referred to by the full term "Erysieplothrix sp. strain 2", which was also misspelt. This can be replaced by the abbreviation once it is established.
Line 76-78: This concluding statement for the introduction was a bit difficult to understand. Please rephrase or break it down into shorter sentences. It may also be more appropriate as the concluding statement in the discussion rather than the introduction.
Materials and methods:
Line 97-99: How was it decided if the micro-organisms were of veterinary interest or not? Please elaborate on the decision criteria here.
Line 103-104: This sentence is confusing. Did you mean that no other Erysipelothrix spp. was found except ES2?
Line 151-152: What did you mean by "most significant distance between farms"? This warrants further explanation.
Discussion:
Line 308-310: This sentence was difficult to understand. Did you mean "and cellular membrane, which were similar to Boerner et al...."? Please clarify.
Line 313-314: This sentence was difficult to understand. Please rephrase.
Line 418: This should be part of the last paragraph and not a new one.
Author Response
Dear Reviewer 1
We are happy with the editor and reviewer of the “Pathogens” for correct our manuscript. We realized that there were some errors during our correction, and we are thankful for all of the reviewer’s observations.
We added the new information (in pink) about a new article accepted for publication in scientific reports that reinforce our results.
Below we have answered reviewer 1
R1. Line 52. First use of abbreviation ES2 but no previous explanation in parentheses.
Response: We inserted the information. We highlight in yellow.
R1. Line 72. Delete extra space in front of ES2.
Response: Ok. We highlight in yellow.
R1. Line 73-74: ES2 was suddenly referred to by the full term "Erysieplothrix sp. strain 2", which was also misspelt. This can be replaced by the abbreviation once it is established.
Response: Ok. We highlight in yellow.
Line 76-78: This concluding statement for the introduction was a bit difficult to understand. Please rephrase or break it down into shorter sentences. It may also be more appropriate as the concluding statement in the discussion rather than the introduction.
Response: We withdraw this sentence because it was not necessary.
Line 97-99: How was it decided if the micro-organisms were of veterinary interest or not? Please elaborate on the decision criteria here.
Response: The new information is in yellow.
Line 103-104: This sentence is confusing. Did you mean that no other Erysipelothrix spp. was found except ES2?
Response. We restructured the phrase. We highlight in yellow.
Line 151-152: What did you mean by "most significant distance between farms"? This warrants further explanation.
Response. We restructured the sentence. We highlight in yellow: “The longer distance among turkey farms was up to 110 km”.
Discussion:
Line 308-310: This sentence was difficult to understand. Did you mean "and cellular membrane, which were similar to Boerner et al...."? Please clarify.
Response. There was an error and missed a period. We changed the phrase and we highlight in yellow.
Line 313-314: This sentence was difficult to understand. Please rephrase.
Response. We changed the sentence and we highlight in yellow
Line 418: This should be part of the last paragraph and not a new one.
Response. We changed it and we highlight in yellow

Reviewer 2 Report
Pathogens-2020 Revision: “First report of epidemiology and genetic variability of Erysipelothrix sp. strain 2 in turkeys associated to Vero cells morphometric alteration”
The aim of the study was to evaluate the epidemiology and genetic variability of Erysipelothrix sp. strain 2 isolates from an outbreak event in infected turkeys and some related pathogenetic mechanisms.
General observations
The study highlights findings in the field of epidemiology and pathogenetic mechanisms of Erysipelothrix in turkeys. Erysipelothirx could be an important issue in farms and for humans, as an important cause of occupational disease. In particular, Authors studied 19 isolates from an outbreak in Brazil, describing a high variability among isolates by molecular typing, with only two clones and two clusters. After performing the morphometry, Authors recorded enlargement of the nucleus and nucleolus, by infecting Vero cells. Moreover, they performed fluorescence assays that resulted in apoptotic and necrotic cells, also demonstrating invasion of Vero cells and multiplication in extracellular medium. These findings could demonstrate that ES2 could have a similar behavior as E. rhusiopathiae as a facultative intracellular microorganism, which may represent a hazard for humans
I would require only some minor points:
- Describe in brief the typical antimicrobial treatment in turkeys to prevent (or to treat) erysipelas.
- Susceptibility to antibiotics should be assessed for the 19 Erysipelothrix sp. strain 2 isolates, since this species could represent an hazard for humans and very few data exist.
- Please, add a dotted line in Figure 2 (at 90% of similarity) in order to define clusters.
Author Response
Dear reviewer 2
We are happy with the editor and reviewer of the “Pathogens” for correct our manuscript. We realized that there were some errors during our correction, and we are thankful for all of the reviewer’s observations.
We added the new information (in pink) about a new article accepted for publication in scientific reports that reinforce our results.
Below we have answered the reviewer.
R2. Describe in brief the typical antimicrobial treatment in turkeys to prevent (or to treat) erysipelas.
Response. We added the information in the discussion highlight in green.
R2. Susceptibility to antibiotics should be assessed for the 19 Erysipelothrix sp. strain 2 isolates, since this species could represent an hazard for humans and very few data exist.
Response. We studied this susceptibility with other isolates during the same outbreak in another article published in 2019 (https://doi.org/10.1007/s13213-019-01505-3). In the present paper, this is not our objective.
R3. Please, add a dotted line in Figure 2 (at 90% of similarity) in order to define clusters.
Response. We pointed the cluster with a black arrow, and we added the new subtitle with the information.
